# Molecular Dynamics-Assisted Design of High Temperature-Resistant Polyacrylamide/Poloxamer Interpenetrating Network Hydrogels

**DOI:** 10.3390/molecules27165326

**Published:** 2022-08-21

**Authors:** Xianwen Song, Gang Lu, Jingxing Wang, Jun Zheng, Shanying Sui, Qiang Li, Yi Zhang

**Affiliations:** 1State Key Laboratory of Shale Oil and Gas Enrichment Mechanism and Effective Development, Beijing 100083, China; 2Research and Development Center for the Sustainable Development of Continental Sandstone Mature Oilfield by National Energy Administration, Beijing 100083, China; 3Hunan Provincial Key Laboratory of Micro & Nano Materials Interface Science, College of Chemistry and Chemical Engineering, Central South University, Changsha 410083, China

**Keywords:** acrylamide polymers, high temperature-resistant, hydrogel, hydrogen bonding, molecular dynamics, poloxamers

## Abstract

Polyacrylamide has promising applications in a wide variety of fields. However, conventional polyacrylamide is prone to hydrolysis and thermal degradation under high temperature conditions, resulting in a decrease in solution viscosity with increasing temperature, which limits its practical effect. Herein, combining molecular dynamics and practical experiments, we explored a facile and fast mixing strategy to enhance the thermal stability of polyacrylamide by adding common poloxamers to form the interpenetrating network hydrogel. The blending model of three synthetic polyacrylamides (cationic, anionic, and nonionic) and poloxamers was first established, and then the interaction process between them was simulated by all-atom molecular dynamics. In the results, it was found that the hydrogen bonding between the amide groups on all polymers and the oxygen-containing groups (ether and hydroxyl groups) on poloxamers is very strong, which may be the key to improve the high temperature resistance of the hydrogel. Subsequent rheological tests also showed that poloxamers can indeed significantly improve the stability and viscosity of nonionic polyacrylamide containing only amide groups at high temperatures and can maintain a high viscosity of 3550 mPa·S at 80 °C. Transmission electron microscopy further showed that the nonionic polyacrylamide/poloxamer mixture further formed an interpenetrating network structure. In addition, the Fourier transform infrared test also proved the existence of strong hydrogen bonding between the two polymers. This work provides a useful idea for improving the properties of polyacrylamide, especially for the design of high temperature materials for physical blending.

## 1. Introduction

Polyacrylamide (**PAM**), a water-soluble material polymerized from acrylamide monomers [1,2], is widely used in many fields, such as oil extraction, metallurgy, papermaking, and medicine [3,4]. Due to its excellent properties such as flocculation, thickening, resistance reduction, and dispersion, **PAM** with high molecular weight can increase the viscosity of oil field injection fluid and improve the heterogeneity of reservoirs, so it is widely used as an oil displacement agent in the process of oil recovery [5,6,7]. However, due to the curl shrinkage of the **PAM** molecular chain and the partial hydrolysis of the amide group on **PAM** at higher temperatures, the viscosity of the polymer decreases significantly with an increase in temperature, which greatly limits its application range. The viscosity drops rapidly, which seriously restricts its application [8,9,10]. Therefore, the design and preparation of high temperature-resistant **PAM** have attracted more and more attention.

In recent decades, scientists have carried out numerous innovative works on improving the temperature resistance of polymers. The research directions mainly include two aspects: 1. Synthesize copolymers with temperature-resistant structural units [11,12,13]. The special properties of polymers obtained by copolymerizing one or more temperature-resistant monomers with acrylamide monomers, such as hydrophobically associative, molecular complex, and zwitterionic [14,15], enable the hydrolysis process under restricted high temperature conditions, thereby improving the stability of the material. 2. Blend with some special substances [16,17,18]. Through chemical interaction (Schiff base, grafting, and substitution reaction) or physical interaction (hydrophobic association, hydrogen bonding, and electrostatic interaction) [18,19], it is easier to form a cross-linked structure between the **PAM** molecules. This can make the polymer stronger and more difficult to transform conformation, thereby improving its stability at high temperature. Compared with the complicated synthesis process, the addition of foreign substances to form the composite system of cross-linked polymers is simpler, and it is also beneficial for large-scale preparation and application. Although the optional additives are also various, such as organic chromium, phenolic, and polyethyleneimine, etc., these materials require caution due to their cytotoxicity and environmental concerns [19,20,21]. Furthermore, developing a class of efficient and safe additives undoubtedly requires more effort and time, which is very challenging for researchers.

Herein, with the assistance of molecular dynamics simulations, we report a strategy to improve the thermal stability of polyacrylamide by using poloxamer as a crosslinker to form an interpenetrating network hydrogel. The so-called poloxamer is a new type of polymer nonionic polyol with low toxicity, low irritation, and good biocompatibility, which can be used in biological fields such as pharmaceutical excipients, gene therapy, and the inhibition of postoperative intestinal adhesions [22,23,24]. First, multiple models were established to simulate the interaction between poloxamers and three common polyacrylamides, which were anionic polyacrylamide (**APAM**) [25], cationic polyacrylamide (**CPAM**) [26], and nonionic polyacrylamide **(NPAM**) [27]. The results predicted strong hydrogen bonds between the amide groups on the polymer chain and the poloxamers. Subsequently, the different viscosity characteristics and morphological structures of the three composite systems at high temperature were verified experimentally. Finally, further rheological tests also indicated that the **NPAM**/poloxamer composite system is a high-viscosity hydrogel with ideal thermal stability and high temperature resistance. This facile mixing method does not require further complicated processing steps, which facilitates large-scale production and the rapid preparation of materials and has broad application prospects in the fields of oil extraction, papermaking, and environmental protection. Furthermore, this work utilizes molecular dynamics as a guide to develop suitable additives for polyacrylamide, providing a useful idea for the design of high temperature materials for physical blending.

## 2. Results and Discussion

### 2.1. Molecular Dynamics

Figure 1 shows the synthetic process and chemical structures of three different types of polyacrylamides, as well as the structures of poloxamers. The structures of **CPAM**, **APAM**, **NPAM,** and poloxamer can be verified in the ^1^H NMR spectrum (Appendix A), which mainly show inactive hydrogens on methyl or methylene groups. To investigate possible interactions between such polymers, we first performed molecular dynamics (MD) simulations of simple **NPAM** and poloxamer [28,29,30]. Nonionic polyacrylamide (**NPAM**) has only -CO-NH_2_ groups and no other ionic groups. The MD processes and interactions between the two polymers were mainly studied. In an aqueous solution, the molar ratio of **NPAM** to poloxamer is 1:1. Initially, 20 molecules were randomly contained in a cube length of 45 Å, equilibrated at 298.15 K for 1 ns, and finally MD simulations of 50 ns were performed. Eventually all molecules form different structures. After the simulation process, the molecular dynamics simulation plots at 10 ns, 35 ns, and 50 ns were extracted, and different colors were used to distinguish the two molecules (Figure 2A). It can also be seen from the simulation results that an aggregated complex is formed between poloxamer and nonionic polyacrylamide, which is mainly due to the obvious hydrogen bonding between the hydroxyl, ether, and amide groups. Note that there are also obvious hydrogen bonds between the **NPAM** molecules. According to reports, multiple hydrogen bonds formed between polymers are beneficial to improving the stability of materials in most cases.

In addition, MD simulations were performed on other composite systems under the same conditions (Appendix A). For anionic polyacrylamide (**APAM**), it has a -COO^−^ group and therefore has a pronounced negative charge (Figure 3A). For cationic polyacrylamide (**CPAM**), it has -N(CH_3_)^3+^ groups in cationic form on the branched chains, which have a pronounced positive charge (Figure 3B). Unlike the **NPAM** case, the binding of **APAM** and **CPAM** to poloxamer was not evident. **APAM** and poloxamer tend to disperse rather than aggregate. This may be due to the fact that the carboxyl groups on **APAM** are easily soluble in water [31,32], that is, tend to combine with water molecules, which instead reduces the complexation with poloxamer. For **CPAM**, there is a large branch on the molecular chain, which may be difficult to disperse due to the fact that the steric position can fix the poloxamer (Figure 3B) molecule [33,34]. From the results of the two simulations, poloxamer molecules still tend to form hydrogen bonds with amide groups on the polymer chain, rather than other anionic and cationic groups on the copolymer. Due to the relatively low content of amide groups in these two copolymers, it may lead to weaker hydrogen bonding with poloxamers.

### 2.2. Preparation and Characterization of Materials

Furthermore, the predicted results of molecular dynamics simulations are verified by actual experiments. For the convenience of the study, the ratio of polyacrylamide and poloxamer was fixed at 1:1, and the total content of both in composite systems was 10 wt%. Typically, the three low-concentration polyacrylamides do not gel at room temperature or high temperature. After adding poloxamer and mixing at room temperature to dissolve, **CPAM**/poloxamer and **NPAM**/poloxamer formed a more obvious self-supporting hydrogel (Figure 4A). In contrast, **APAM**/poloxamer was in the solution state. Figure 4B and Figure 4C also provide the G′ (storage modulus) and G″ (loss modulus) of **CPAM**/poloxamer and **NPAM**/poloxamer. Except for **APAM**/poloxamer (Appendix A), the G′ values of the other blends were consistently greater than G″ over the entire range, indicating the elastic properties of the hydrogels [35]. In addition, it can also be found that the G′ value of **NPAM**/poloxamer (135 Pa) was higher than that of **CPAM**/poloxamer (98 Pa), which means higher intensity [36,37].

The effect of temperature on the viscosity of the three composite systems was also investigated. Meanwhile, three single polymer solutions with a concentration of 10 wt% were also prepared for comparison. According to reports, poloxamer is a classical temperature-sensitive copolymer, which produces a sol-gel phase transition with the change of temperature [22]. When the temperature increases, its viscosity first increases with the temperature and then slowly decreases (Appendix A). The viscosity of **CPAM**/poloxamer is relatively stable, which may benefit from the reduced temperature sensitivity due to the introduction of large cationic groups (Figure 5A). The viscosity of **NPAM**/poloxamer is also more stable over the entire temperature range (25–80 °C) compared to pure **NPAM** (Figure 5B). When the temperature was raised to 70 °C, the viscosity of the former dropped sharply to 12,800 mPa·S, but the latter remained at 25,150 mPa·S. Therefore, it can be pointed out that poloxamer significantly improves the thermal stability of the viscosity of **NPAM** at high temperature. As shown in Figure 5C, the addition of poloxamer did not significantly change the viscosity of **APAM** at high temperature, but instead caused a sharp drop. According to reports, it is known that poloxamer molecules are negatively charged due to the presence of ether groups (-O-) and hydroxyl groups (-OH). This leads to mutual repulsion with the similarly negatively charged **APAM** molecules, making it difficult for the two to bind and possibly reducing the viscosity of the latter.

The shear viscosities of the three complexes were also tested at continuously varying shear rates. In theory, composites tend to have higher viscosities than single polymers because crosslinking can restrict the movement of polymer chains. The viscosity of **CPAM**/poloxamer did not change much (Appendix A), and a very significant increase in the viscosity of **NPAM**/poloxamer was noted over the entire shear range (Appendix A). In addition, compared to the single **APAM** solution, the viscosity of **APAM**/poloxamer dropped sharply (Appendix A). The viscosity of all materials decreased with an increasing shear rate, indicating that they have polymer properties [38,39].

Furthermore, SEM was used to study the differences between the three complexes. Hydrogels typically contain a regular three-dimensional (3D) framework and water. After freeze-drying to remove moisture, the pore structure in the framework can be observed by SEM [40,41]. As shown in Figure 5D, the shape of the **CPAM**/poloxamer sample has irregular pore walls and loose pores which are about 3 µm, while the shape of the **NPAM**/poloxamer sample has a dense and regular pore structure (around 1 µm). These observations suggest that at the beginning of **NPAM,** molecules and poloxamer chains cross-link with each other and form an interpenetrating network (Figure 5E) [10,42,43]. Apparently, the SEM image of **APAM**/poloxamer does not show obvious micropores (Figure 5F), but just some scattered and irregular flakes. It has been reported that the complete interpenetrating network structure is beneficial to the stability of the material, which may be the reason for the high temperature resistance of **NPAM**/poloxamer.

### 2.3. The Formation Mechanism of Hydrogels

Due to the excellent thermal stability of **NPAM**/poloxamer hydrogel, a more comprehensive characterization was performed. In fact, the hydrogel still had some fluidity at low concentration (4 wt%), while at high concentration (12 wt%) it was easy to generate bubbles and led to opacity (Figure 6A). It is speculated that the optimal concentration of the composite system may be between 6 and11 wt%. For convenience, **NPAM**/poloxamer-4, **NPAM**/poloxamer-8, and **NPAM**/poloxamer-12 were used to represent the hydrogels at the concentrations of 4, 8, and 12 wt%, respectively. To further elucidate the interactions of mixtures, Fourier transform infrared spectroscopy (FTIR) was performed. As can be seen in Figure 6B, the characteristic O-H and C-O stretching bands of poloxamer are shown at 3450 and 1105 cm^−1^, respectively [44,45,46]. After blending with **NPAM**, the characteristic stretching bands of poloxamer shifted to 3445 and 1100 cm^−1^, respectively, and a distinct broad peak was formed in the region near O-H. As the concentration of **NPAM**/poloxamer increases, the characteristic peak shifts of O-H and C-O are more obvious. The shift in the FTIR spectrum towards lower wavenumbers is reported to demonstrate the formation of stronger hydrogen bonds [46,47,48].

In addition, hydrogel concentrations may affect the structure of the composite system, and SEM was used to study the composite system at different concentrations. As shown in Figure 6C, the lyophilized **NPAM**/poloxamer-4 sample has an irregular multi-layered scale structure with insignificant micropores. With increasing concentration, the **NPAM**/poloxamer-8 hydrogel showed continuous pore walls and regular micropore, indicating the formation of a denser interpenetrating network structure. Similarly, **NPAM**/poloxamer-12 hydrogel has a distinct interpenetrating network structure, but also has a ragged pore structure which may be caused by inhomogeneous mixing. These results provide additional evidence for the formation of hydrogels and also demonstrate a stronger interaction between poloxamer and **NPAM**. Furthermore, this seems to validate the results of the above molecular dynamics simulations where the amide groups(-CO-NH_2_) of **NPAM** have strong interactions with the ether groups (-O-) and (-OH) of poloxamer, mainly in the form of hydrogen bonds.

### 2.4. Thermosensitivity of Hydrogels

Since the hydrogen bonding between the poloxamer and the nonionic polyacrylamide is crucial for gel formation, the concentration has a crucial effect on the performance of the hydrogel. Therefore, we investigated the stability of hydrogels with different concentrations at high temperature. Figure 7A is the change curve of the three concentrations of the **NPAM**/poloxamer composite system with the increase of temperature. It can be seen from the figure that with an increase in concentration, the viscosity of the composite system tends to be stable in the process of increasing the temperature. For **NPAM**/poloxamer-4 hydrogel, the viscosity is stable between 8149–32,500 mPa·S in the range of 50–80 °C, and the variation range is relatively large. Obviously, the viscosity of the hydrogel further increased with increasing concentration. Note that at 75 °C, the viscosity of the **NPAM**/poloxamer-4 hydrogel is 11,000 mPa·S compared to 23,700 mPa·S for **NPAM**/poloxamer-8, which is an improvement of approximately two-fold. However, the viscosity of **NPAM**/poloxamer-12 is only 28,200 mPa·S at the same temperature, and there is no obvious increase. This shows that after the concentration reaches a certain level, the maximum viscosity of the hydrogel at high temperature has reached saturation.

The temperature sweep data of G′ (storage modulus) and G″ (loss modulus) of the composite system were further investigated. As shown in Figure 7B, although the G′ of the **NPAM**/poloxamer-4 hydrogel is always higher than G″, the decreasing trend is very obvious, which indicates the instability of the hydrogel structure [30,49]. In addition, the G′ of **NPAM**/poloxamer-8 (Figure 7C) and **NPAM**/poloxamer-12 (Figure 7D) showed similar variation patterns in the whole temperature range, and the magnitude of decrease was much smaller. In addition, it can also be seen from the data that the viscosity of all hydrogels shows a slow increase between 25 and 35 °C, and then decreases. In the early stage of the temperature increase, the poloxamer gelled and caused the viscosity to increase (Appendix A). This might also enhance the hydrogen bonding with **NPAM** molecules, thereby improving the stability of the solution at high temperature. The stabilization of the hydrogel may also benefit from the complete interpenetrating network formed by multiple hydrogen bonds which can greatly enhance the strength of the structure [50,51]. As the temperature continued to increase, the hydrogen bond between the polyacrylamide and poloxamer in the composite system would become weaker, which eventually led to a decrease in viscosity.

It is worth noting that high temperature also has an effect on the appearance of hydrogels. It can be seen that **NPAM**/poloxamer-8 hydrogel appeared with obvious turbidity at 80 °C and gradually clarified after lowering the temperature (Figure 7E). When returned to a room temperature of 25 °C, the composite system returned to its original clear and transparent state. It has been reported that poloxamer will appear cloudy and gelatinized at high temperature [52,53], which may be beneficial to the stabilization of **NPAM**/poloxamer. Therefore, this facilely prepared blended hydrogel is expected to be used to improve the application range of polyacrylamide, especially in high temperature environments.

## 3. Materials and Methods

### 3.1. All-Atom Molecular Dynamics (AAMD)

Fragment structures of poloxamers and three polyacrylamides (**NPAM**, **CPAM**, and **APAM**) were established by Guassian16. Afterwards, geometry optimization was performed by the B3LYP-D3 method and 6–31G (d, p) basis set. Atomic and molecular dynamics simulations were performed in the GROMACS (version 2020.6) simulation package using the general Amber force field (GAFF) combined with the TIP3P water model [54]. The ratio of poloxamers to **NPAM** was 1:1, and all systems were constructed by randomly placing 50 molecules into cubic boxes of approximately 6 nm. After solvation with water molecules, molecular dynamics simulations in the isothermal and isobaric (NPT) ensemble after thousands of steps of energy minimization were performed using Berendsen’s method at 298 K, 1 atm, and 50 ns. Structural pictures at 0 ns, 10 ns, 35 ns, and 50 ns were extracted to observe the interactions between molecules. The cutoff length for non-bonded interactions is 1.2 nm, and the long-range electrostatic interactions use the particle grid Ewald method with a Fourier spacing of 0.1 nm. Constraint of all covalent bonds to hydrogen atoms used the LINCS algorithm. **CPAM** and **APAM** were simulated using the same method. The final result was displayed by VMD 1.9.3 software [55], while the molecular structure was displayed by open-source PYMOL 2.3.0.

### 3.2. Materials

Acrylamide (A.R, 99.0%), acrylic acid (A.R, 99%), acryloyloxyethyltrimethyl ammonium chloride (A.R, 99%), poloxamer (A.R.), and other chemicals were purchased from Aladdin Reagent Co., Ltd. (Shanghai, China). All chemicals were used without purification, and deionized water was used for all experiments.

### 3.3. Synthesis of Nonionic Polyacrylamide (NPAM)

To begin with, 2 g of acrylamide was added to enough deionized water, stirred slowly until the acrylamide was completely dissolved. The pH was adjusted to about 4, stirred again and passed through N_2_ gas to remove air. Subsequently, amine persulfate-sodium bisulfite was added as an initiator, and the resulting mixed solution was reacted at 25 °C for 24 h. The product was then precipitated with absolute ethanol and washed 3 times. Finally, a white powdery **NPAM** sample was obtained by freeze-drying for 24 h.

### 3.4. Synthesis of Anionic Polyacrylamide (APAM)

A mixed solution prepared from 2 g of acrylamide and 2 g of acrylic acid was adjusted to pH 9.0 with NaOH, and N_2_ gas was introduced to remove air. The next steps were as above.

### 3.5. Synthesis of Cationic Polyacrylamide (NPAM)

A mixed solution of 2 g of acrylamide and 2 g of acryloyloxyethyltrimethyl ammonium chloride was prepared, and N_2_ gas was introduced to remove air. The next steps were as above.

### 3.6. Preparation of Hydrogels

The obtained polymers were freeze-dried to obtain powder products. Poloxamer solution (5 wt%) was obtained by dissolving poloxamer powder into deionized water at a temperature of 50 °C for several minutes. Then, nonionic polyacrylamide (5 wt%) was added and stirring was continued for a few minutes to finally obtain a 10 wt% **NPAM**/poloxamer mixture. After sonication to remove air bubbles, the product was freeze-dried to obtain a powder. Cationic polyacrylamide and anionic polyacrylamide were prepared using the same method.

### 3.7. Characterization

^1^H NMR spectra were recorded by using an AMX-400 (Bruker, Switzerland). The freeze-dried samples were dissolved in D_2_O and scanned under a temperature of 300 K. The microstructure images of **NPAM**/poloxamer hydrogels were characterized by a scanning electron microscope (SEM, JEOL JSM-6701F, Japan, Tokyo). FT-IR spectra of samples were obtained with a Perkin Elmer Spectrum One instrument (Waltham, MA, USA) and the scanning range was 4000 cm^−1^–400 cm^−1^. The test samples were prepared by mixing freeze-dried hydrogel powder after 48 h of vacuum drying with KBr crystals baked by infrared light.

### 3.8. Rheological Mechanical

The rheological behavior of **NPAM**/poloxamer samples was analyzed using a modular compact rheometer (Anton Paar, MCR 302). Time sweep measurements were performed at 0.1% strain, 1 Hz frequency, and the time was held at 20 min. Frequency (ω) sweep tests were performed in the range 0.1–100 rad/s and the temperature was fixed at 25 °C. The temperature sweep test (T) was performed in the range of 25–80 °C with ω fixed at 0.5 rad/s. The strain (γ) was fixed at 0.5% in all the above tests.

## 4. Conclusions

In this study, a polyacrylamide/poloxamer hydrogel with an interpenetrating network was successfully constructed using a hybrid modification strategy. To study the interactions between polymers, molecular simulations were used to study the formation of three polyacrylamides and poloxamers, which were anionic polyacrylamide, cationic polyacrylamide, and nonionic polyacrylamide. The simulation results predict strong hydrogen bonding between the amide group on the polyacrylamide and the oxygen-containing groups (ether and hydroxyl) on the poloxamer, which may be the point of maintaining stability at high temperature.

Subsequent experiments tested the properties of the three complexes and verified the effect of poloxamers on polyacrylamides within different groups. The results of this experiment also showed that the addition of poloxamers significantly increased the viscosity and stability at high temperature of the nonionic polyacrylamide. Through a series of characterizations, it was concluded that the prepared nonionic polyacrylamide/poloxamer composite system formed an interpenetrating three-dimensional network with a regular porous structure which could form a self-supporting hydrogel at room temperature. In addition, the experiments further proved that when the ratio of the two polymers was 1:1, the hydrogel with a content of 8 wt% already had excellent high temperature resistance properties.

Considering the results obtained in this study, it can be emphasized that nonionic polyacrylamides based on non-toxic and harmless poloxamers are very suitable for use as additives, resulting in higher viscosity and more remarkable stability. Since this material is easy to prepare and suitable for large-scale production, it has great application potential in petroleum, water treatment, and other fields. Furthermore, the introduction of molecular dynamics simulations also provides a simple and practical strategy for designing high temperature-resistant acrylamide polymers.

## Figures and Tables

**Figure 1 molecules-27-05326-f001:**
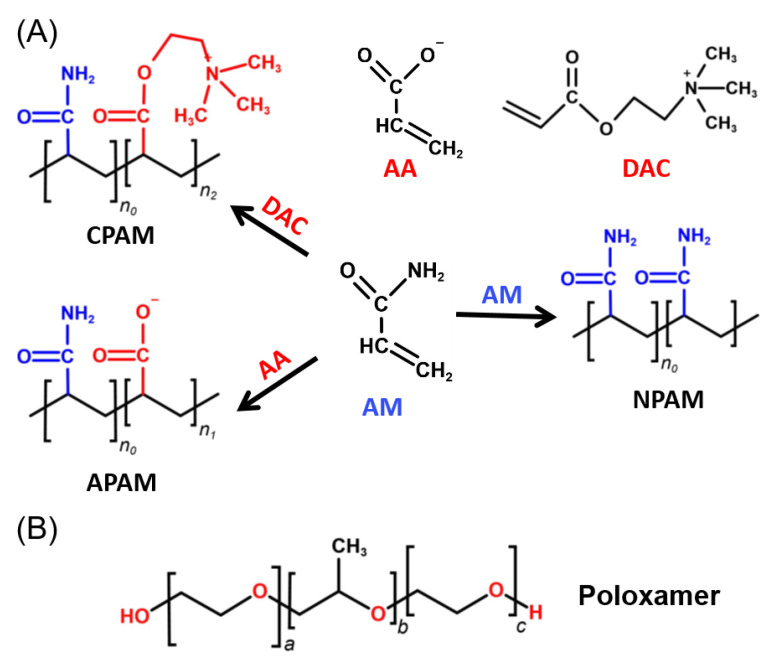
(**A**) Structure of anionic polyacrylamide (**APAM**), cationic polyacrylamide (**CPAM**), and nonionic polyacrylamide (**NPAM**). (**B**) Structure of Poloxamer. Among them, important monomers include acrylamide (AM), acrylic acid (AA), and acryloyloxyethyltrimethyl ammonium chloride (DAC).

**Figure 2 molecules-27-05326-f002:**
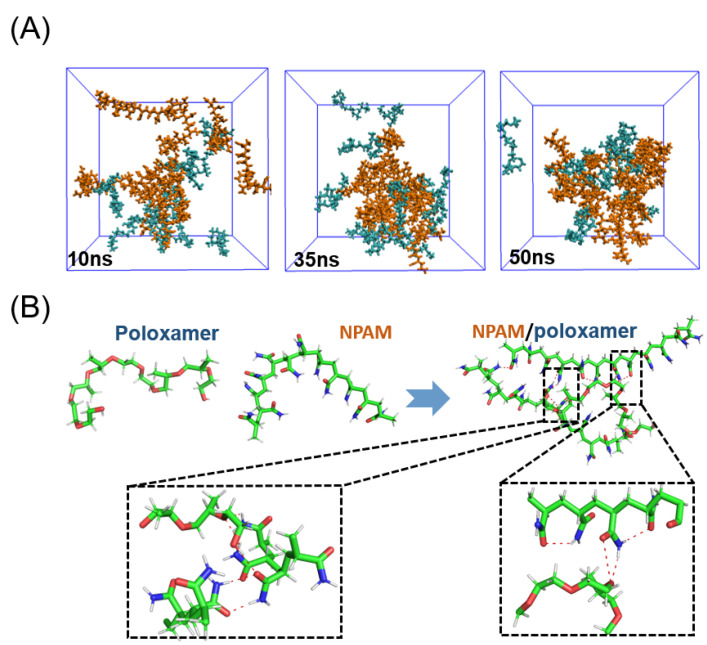
(**A**) Molecular dynamics process of the interaction between **NPAM** and poloxamer. (**B**) Hydrogen bonding between two polymers.

**Figure 3 molecules-27-05326-f003:**
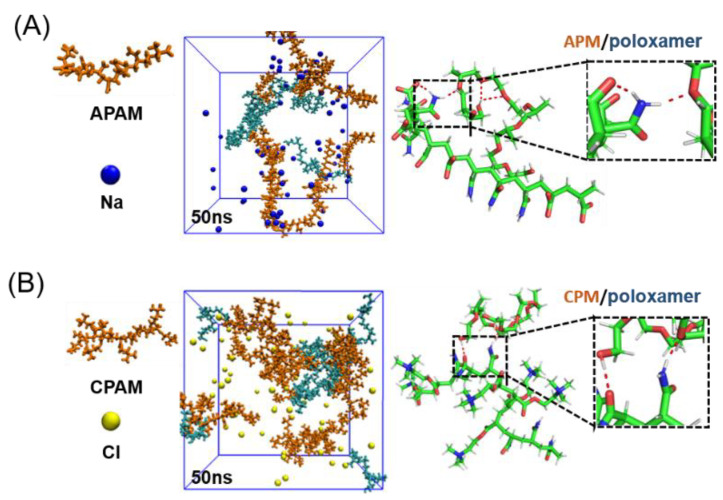
Molecular dynamics results of the interaction of two composite systems. (**A**) **APAM** and poloxamer. (**B**) **CPAM** and poloxamer.

**Figure 4 molecules-27-05326-f004:**
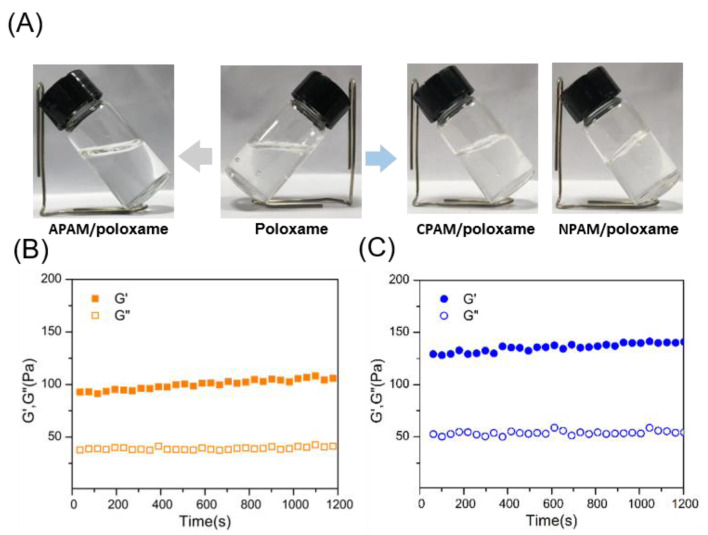
(**A**) Pictures of poloxamer, **APAM**/poloxamer, **CPAM**/poloxamer, and **NPAM**/poloxamer. G′ and G″ values of (**B**) CPAM/poloxamer and (**C**) NPAM/poloxamer.

**Figure 5 molecules-27-05326-f005:**
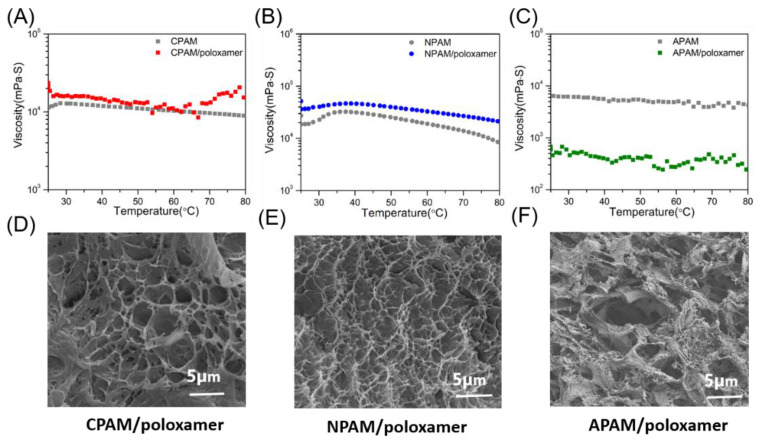
Viscosity of (**A**) **CPAM** and **CPAM**/poloxamer, (**B**) **NPAM** and **NPAM**/poloxamer, and (**C**) **APAM** and **APAM**/poloxamer. The temperature range is 25–80 °C. (**D**–**F**) SEM images of the three composite systems. All concentrations are 10 wt%.

**Figure 6 molecules-27-05326-f006:**
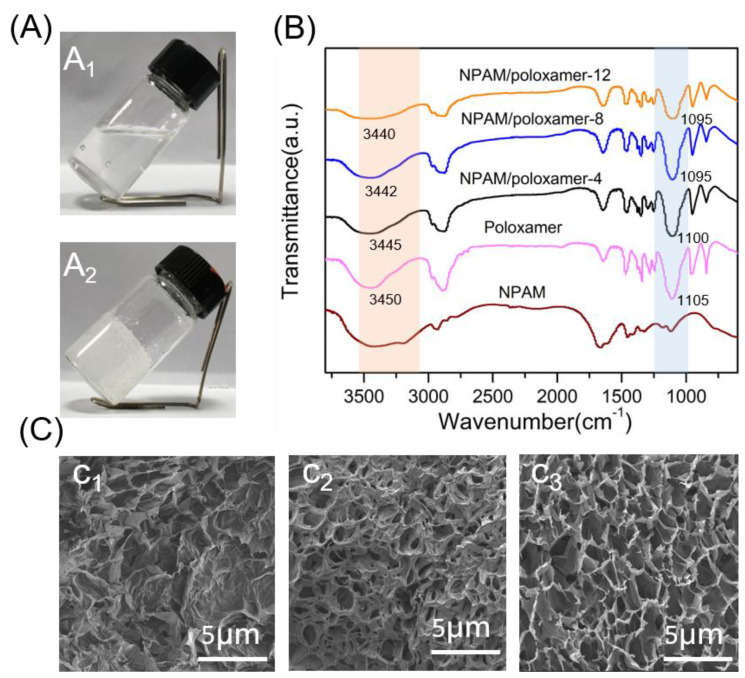
(**A**) Digital images of **NPAM**/poloxamer-4 and **NPAM**/poloxamer-12 hydrogels. (**B**) FTIR of **NPAM**, poloxamer, **NPAM**/poloxamer-4, **NPAM**/poloxamer-8, and **NPAM**/poloxamer-12. (**C**) SEM images of three hydrogels at different concentrations.

**Figure 7 molecules-27-05326-f007:**
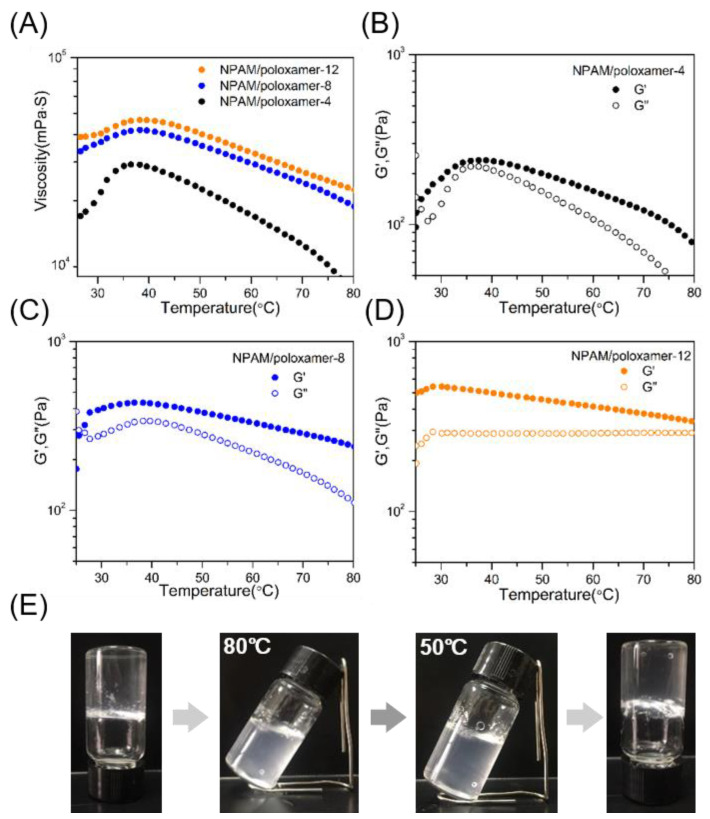
(**A**) Viscosity of **NPAM**/poloxamer hydrogels at different concentrations. Temperature sweep tests of (**B**) **NPAM**/poloxamer-4, (**C**) **NPAM**/poloxamer-8, and (**D**) **NPAM**/poloxamer-12 hydrogels. (**E**) Pictures of **NPAM**/poloxamer-8 hydrogel heated to 80 °C and cooled to 25 °C.

## Data Availability

The data presented in this study are available in Appendix A.

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
