# Peer review of "Molecular Dynamics-Assisted Design of High Temperature-Resistant Polyacrylamide/Poloxamer Interpenetrating Network Hydrogels"

_molecules, 2022, doi:10.3390/molecules27165326_

Round 1

Reviewer 1 Report

The manuscript entitled "Molecular Dynamics-assisted Design of High Temperature-Resistant Polyacrylamide/Poloxamer Interpenetrating Network Hydrogels" reports novel and interesting data and results.

Nevertheless, the manuscript needs a minor revision before its publication in the Journal Molecules. In more details: Authors should: (a) proceed to an extensive editing of English language, of their text; (b) rewrite all of their methods by just describing the followed experimentation procedure (step-by-step) and not by giving "orders", as in the present version; (c) improve the section Conclusions appropriately.

Author Response

Point-by-point response to the individual questions of Reviewer 1:

Over all Comment:

The manuscript entitled "Molecular Dynamics-assisted Design of High Temperature-Resistant Polyacrylamide/Poloxamer Interpenetrating Network Hydrogels" reports novel and interesting data and results. Nevertheless, the manuscript needs a minor revision before its publication in the Journal Molecules. In more details: Authors should:

Comment 1: proceed to an extensive editing of English language, of their text.

Response: Thank you very much for promptly pointing out the language problems with this paper. Based on your suggestion, we have rechecked the presentation of the full text to describe our work as clearly as possible. The revised paper is more coherent and easier to understand, which will facilitate the reading of a wide range of readers.

Comment 2: Rewrite all of their methods by just describing the followed experimentation procedure (step-by-step) and not by giving "orders", as in the present version.

Response: Thank you for pointing out this error. It's true that the descriptions in our original manuscript were overly lengthy and unsuccinct, which may not be easy to read. According to your comments, we re-describe the step-by-step experimental procedure for the preparation of polymers such as polyacrylamide. The corrected part is on Page 9, Lines 27-35 of the main manuscript.

Comment 3: Improve the section Conclusions appropriately.  

Response: Thanks for your important suggestion. We re-examined the conclusion section in the manuscript and found that there are indeed deficiencies as you stated. After referencing other published articles in this journal, we have made significant revisions to the conclusions and are more rigorous and detailed. Thanks again for your suggestion. The corrected part is on Page 10, Lines 15-40 of the main manuscript.

Reviewer 2 Report

1) Please remove acronyms from the abstract

2) keywords should be in alphabetic order

3) The number of acronyms should be reduced in the manuscript's text to make it readable.

4) The standard structure of the article is recommended - Introduction - Materials and Methods - Results - Discussion - Conclusions

5) The first sentence of the abstract has no relation to the abstract but suits well to the introduction.

6) "exhibiting special high temperature-resistant properties" - please add data on temperature resistance.

7) Abstract should also describe the used methods and techniques.

8) " a simple and effective strategy" - what were your criterium for simplicity and efficiency? What were alternative strategies used for comparison?

9) "which is non-toxic, harmless, safe and environmentally friendly, and is widely used in the biological field" - please add countable data

10) Please add a countable study goal at the end of the introduction, formulate the novelty of the work, and add your study tasks (1-2-3-4).

11) The conclusions should be countable. 

12) Please add the future outlook and possible impact of your findings on the industry.

13) please avoid personal sentences such as "we ..."

Author Response

Point-by-point response to the individual questions of Reviewer 2:

Over all Comment:

Comment 1: Please remove acronyms from the abstract.

Response: Thank you for your reminder. Following your suggestion, we have reworked the abstract section of the manuscript, omitting the original abbreviations. Indeed, this will make the abstract more concise and readable.The corrected part is on Page 1, Lines 31 of the main manuscript.

Comment 2: Keywords should be in alphabetic order

Response: Thank you very much for your careful reading. In alphabetical order, we have reworked the keywords in the manuscript. The corrected part is on Page 1, Lines 31 of the main manuscript.

Comment 3: The number of acronyms should be reduced in the manuscript's text to make it readable.  

Response: Thank you very much for your valuable suggestions. After re-examination of the original manuscript, it was found that the large number of acronyms made a considerable part of the content difficult to understand, causing confusion in reading. Based on your comments, we have made extensive revisions to the full text, including text and images. According to the consensus of previous literature, only common words such as "nonionic polyacrylamide" are abbreviated as NPAM, but "poloxamer" is retained. In addition, the previous excessive abbreviations such as "NPL" were also abandoned, so as not to cause reading difficulties for readers.The revised article will be more conducive to readers' reading.

Comment 4: The standard structure of the article is recommended - Introduction - Materials and Methods - Results - Discussion – Conclusions

Response: Thank you for your reminder. Yes, your opinion is correct. After we carefully refer to the many papers published in this journal, there seems to be a default order, which is "Introduction - Results and Discussion-Materials and Methods - Conclusions". In addition, the template files provided by the journal are in this order. This order of writing may have little effect on the reader. In order to keep the format of the article consistent, it seems more appropriate to keep the original writing order after comprehensive consideration. Thanks again for your careful reading.

Comment 5: The first sentence of the abstract has no relation to the abstract but suits well to the introduction.

Response: Thank you for your reminder. Indeed, the first sentence of the abstract describes too many words, and is still not concise and decent enough. Based on your suggestion, we have revised the abstract. The corrected part is on Page 1, Lines 13 of the main manuscript.

Comment 6:wthw "exhibiting special high temperature-resistant properties" - please add data on temperature resistance.

Response: Thank you very much for this very important reminder. BFollowing your suggestion, the Abstract section of the article has added specifics to this paragraph highlighting that the gel still has a high viscosity of 3550 mPa·S at 80°C.  The corrected part is on Page 1, Lines 26 of the main manuscript.

Comment 7: Abstract should also describe the used methods and techniques.

Response: Thank you very much for your valuable suggestions. After rechecking the abstract of the article, it was found that a considerable part of the original content was insufficient and problematic. After referencing numerous literatures, we have significantly revised the abstract of the article, adding descriptions of methods and techniques. We apologize for making more careful revisions prior to submission, and thank you again for your careful reading and these suggestions. The corrected part is on Page 1, Lines 24-31 of the main manuscript.

Comment 8: "a simple and effective strategy" - what were your criterium for simplicity and efficiency? What were alternative strategies used for comparison?

Response: Thank you for pointing out our shortcomings. We restate the advantages of polyacrylamide/poloxamer hydrogels compared to other materials. The point is that the former can be easily prepared with simple mixing, without heating or involving chemical cross-linking, etc. In addition, poloxamers are proven to be an extremely biocompatible, non-polluting substance compared to additives that are harmful to organisms (such as formaldehyde, acetone, etc.). The corrected part is on Page 1, Lines 29 of the main manuscript.

Comment 9: "which is non-toxic, harmless, safe and environmentally friendly, and is widely used in the biological field" - please add countable data

Response: Thank you for pointing out our oversight. As mentioned before, we have added a description of poloxamers and listed their areas of application. The corrected part is on Page 2, Lines 24-27 of the main manuscript.

Comment 10: Please add a countable study goal at the end of the introduction, formulate the novelty of the work, and add your study tasks (1-2-3-4).

Response: Thank you very much for your helpful suggestions, and we apologies for the inadequacy of the introduction. Yes, you are right. A qualified introduction should have the necessary content of the research objectives, the novelty of the work, and the research task. Based on your suggestion, we have revised the introduction to make it clearer and more informative. Thanks again, your suggestion is very helpful for us to improve the quality of the paper. The corrected part is on Page 2, Lines 36-42 of the main manuscript.

Comment 11: The conclusions should be countable.

Response: Thanks for your important suggestion. We re-examined the conclusion section in the manuscript and found that there are indeed deficiencies as you stated. After referencing other published articles in this journal, we have made significant revisions to the conclusions and are more rigorous and detailed. Thanks again for your suggestion.  The corrected part is on Page 10, Lines 15-40 of the main manuscript.

Comment 12: Please add the future outlook and possible impact of your findings on the industry.

Response: Thank you very much for your constructive comments. Based on your suggestion, we have re-examined the presentation of the full text to describe our work as clearly as possible. The revised paper is supplemented with future outlook and possible implications for the industry in the conclusion section. The corrected part is on Page 10, Lines 15-40 of the main manuscript.

Comment 13: please avoid personal sentences such as "we ...".

Response: Thank you very much for your valuable suggestions. After re-checking the manuscript, we also noticed that the word "we" appeared too much. Therefore, the revised manuscript has reorganized the language to improve its scientific nature. Thanks again, your valuable suggestions help us a lot to further improve the quality of the paper.

Reviewer 3 Report

The authors described a physically crosslinked (H-bonding) hydrogel by bending polyacrylamide with poloxmer. They used molecular dynamic to compare the different binding ability of poloxmer with nonionci, anionic, and cationic polyacrylamide and did the experiements to verify the conclusion from the MD study. Overall this manuscript presented a complete scientific study. However, the scientific presentation in this manuscript need to be much improved and some critical experiemtns need to be done before considering acceptance.

1. The synthesis of three different polyacrylamide need to more detailed described. The authors presented the NMR spectra of the synthesized polyacrylamide but no peak assignments in these spectra. The authors need to point out at least the basci peak assignments in the spectra. Another basic synthetic parameter of polymer is the molecular weight and molecular weight distribution. What are the molecular weights and molecular weight distributions of the three synthersized polyacrylamide? Different molecular weights of polymer will of course lead to different viscosity in solution and viscoelastic propeerties. The authors need to demonstrate the three synthesized polyacrylamide have similiar degree of polymerization and PDI by GPC.

2.  In figure 4, the legend for G'' is missing, namely, the empty squre and circle. The same issue in figure S7 and Figure 7B. The authors should pay a lot attention to accurate scientific presentation.

3. In figure 5, the authors try to compare the viscosity of the polyacylamide and mixture of polyacrylamide and poloxamer. The authors should clearly indicate their concentrations. 5% or 10%?In my opinion, the autors should at least compare 3 sample, for example for Figure 5A,  CPM-10%, CPM-POL 10%, POL-10%.

4. In Figure 5c, the addition of POL leads to sharp drop of viscosity. It's not a common phenomonen since higher concentration of polymer will usually have higher viscosity. The authoers need to expain this observation in figure 5c.

5.  In Fiure 7, the authors investigated the thermal stability of different NPL hydrogel.  It's resonlble to observe the decreasing of the viscosity or moduls with the increasing of temperature in the range of 45C to 80C since the H-Bonding will be weaker at higher temperature. However, why there is a peak value in the around 38C? Is it because a better H-bonded hydrogel network formed at this temperature? The authors need to discuss this observation in Figure7.

Author Response

Point-by-point response to the individual questions of Reviewer 3:

Over all Comment:

The authors described a physically crosslinked (H-bonding) hydrogel by bending polyacrylamide with poloxmer. They used molecular dynamic to compare the different binding ability of poloxmer with nonionci, anionic, and cationic polyacrylamide and did the experiements to verify the conclusion from the MD study. Overall this manuscript presented a complete scientific study. However, the scientific presentation in this manuscript need to be much improved and some critical experiments need to be done before considering acceptance.

Comment 1: The synthesis of three different polyacrylamide need to more detailed described. The authors presented the NMR spectra of the synthesized polyacrylamide but no peak assignments in these spectra. The authors need to point out at least the basci peak assignments in the spectra. Another basic synthetic parameter of polymer is the molecular weight and molecular weight distribution. What are the molecular weights and molecular weight distributions of the three synthersized polyacrylamide? Different molecular weights of polymer will of course lead to different viscosity in solution and viscoelastic propeerties. The authors need to demonstrate the three synthesized polyacrylamide have similiar degree of polymerization and PDI by GPC.

Response: Thank you very much for your helpful advice. We apologize for this oversight. According to your suggestion, the H NMR data of the revised paper have all marked the positions of the various peaks, making them clearer and more informative. After multiple inspections, the fundamental peaks in the NMR spectra of the three polyacrylamides have been reasonably assigned, which can be found in Supporting Information (Figure S1, Figure S2, Figure S3 and Figure S4). We totally agree with you that different molecular weight polymers result in different solution viscosity and viscoelasticity. Unfortunately, after waiting almost 10 days, the lab told us that the sample could not be tested, possibly because of the polar groups on the polymer. We are very sorry for not being able to supplement this data. In this experiment, however, the effect of poloxamers on polyacrylamide derivatives with different branches was more investigated, and the viscosity between pure and complex systems was compared. Thanks again, your suggestion is very helpful for us to improve the quality of the paper. The corrected part is on Pages 3-6 of the supporting information.

Comment 2: In figure 4, the legend for G'' is missing, namely, the empty squre and circle. The same issue in figure S7 and Figure 7B. The authors should pay a lot attention to accurate scientific presentation.

Response: Thank you for pointing out our inadequacies. We're sorry for this low-level error in the original manuscript. This problem has been fully corrected in the revised manuscript after re-examination of the content and images. The modified pictures also include Figure 4, Figure 7, and Figure S7-S9.  The corrected part is on Page 5 of the main manuscript.

Comment 3:  In figure 5, the authors try to compare the viscosity of the polyacylamide and mixture of polyacrylamide and poloxamer. The authors should clearly indicate their concentrations. 5% or 10%?In my opinion, the autors should at least compare 3 sample, for example for Figure 5A, CPM-10%, CPM-POL 10%, POL-10%。

Response: Thanks for your very useful advice. Yes, you are right. No attention was paid to these details in the original experimental design. The mass concentrations of all solutions are re-indicated in the revised manuscript, 10 wt% for both single polymer and composite systems. At the same time, the test of 10wt% poloxamer was also added (Figure S8). The description has been re-supplemented, and potentially confusing parts have been removed. . The corrected part is on Page 6 of the main manuscript.

Comment 4: In Figure 5c, the addition of POL leads to sharp drop of viscosity. It's not a common phenomonen since higher concentration of polymer will usually have higher viscosity. The authoers need to expain this observation in figure 5c.

Response: Thanks for your careful reading. In our experiments, we did test that the viscosity of anionic polyacrylamide was greatly reduced after adding poloxamer. After retesting and consulting relevant literature, it can be known that poloxamer has a negative charge due to the presence of ether groups (-O-) and hydroxyl groups (-OH). This mutually repels the also negatively charged anionic polyacrylamide, thus resulting in a reduced viscosity of the latter. We supplement this in this chapter. . The corrected part is on Page 5, Lines 15-20 of the main manuscript.

Comment 5:  In Fiure 7, the authors investigated the thermal stability of different NPL hydrogel.  It's resonlble to observe the decreasing of the viscosity or moduls with the increasing of temperature in the range of 45C to 80C since the H-Bonding will be weaker at higher temperature. However, why there is a peak value in the around 38C? Is it because a better H-bonded hydrogel network formed at this temperature? The authors need to discuss this observation in Figure7.

Response: Thanks for this constructive question. In the experiment, the viscosity of the composite system will gradually increase at 25-40 degrees, and there will be a peak. This phenomenon may be related to the gelation of the added poloxamer. According to Molecules 2022, 27, 299-307 and Abstr Pap Am Chem S 2009, 237, poloxamers are classical temperature-sensitive copolymers that produce sol-gel phase transitions with increasing temperature. In the early stage of the temperature increase, the poloxamer gelled and caused the viscosity to increase (Figure S12). However, as the temperature continued to increase, the hydrogen bond between the polyacrylamide and poloxamer in the composite system would become weaker, which eventually led to a decrease in viscosity. Thanks again for your suggestion, we have added this discussion to the content of the paper.  The corrected part is on Page 8, Lines 2-8 of the main manuscript.

Round 2

Reviewer 2 Report

The authors have ignored most of the comments. The structure of the article stays under the required level.

Reviewer 3 Report

Comments are well addressed.